# Chaotic Regularization and Heavy-Tailed Limits for Deterministic Gradient Descent

**Soon Hoe Lim**
Nordita
KTH Royal Institute of Technology and Stockholm University
`soon.hoe.lim@su.se`

**Yijun Wan**
Département de Mathématiques et Applications
École Normale Supérieure, Université PSL
`wan@clipper.ens.fr`

**Umut Şimşekli**
DI ENS, École Normale Supérieure, Université PSL, CNRS, INRIA
`umut.simsekli@inria.fr`

## Abstract

Recent studies have shown that gradient descent (GD) can achieve improved generalization when its dynamics exhibits a chaotic behavior. However, to obtain the desired effect, the step-size should be chosen sufficiently large, a task which is problem dependent and can be difficult in practice. In this study, we incorporate a chaotic component to GD in a controlled manner, and introduce *multiscale perturbed GD* (MPGD), a novel optimization framework where the GD recursion is augmented with chaotic perturbations that evolve via an independent dynamical system. We analyze MPGD from three different angles: (i) By building up on recent advances in rough paths theory, we show that, under appropriate assumptions, as the step-size decreases, the MPGD recursion converges weakly to a stochastic differential equation (SDE) driven by a heavy-tailed Lévy-stable process. (ii) By making connections to recently developed generalization bounds for heavy-tailed processes, we derive a generalization bound for the limiting SDE and relate the worst-case generalization error over the trajectories of the process to the parameters of MPGD. (iii) We analyze the implicit regularization effect brought by the dynamical regularization and show that, in the weak perturbation regime, MPGD introduces terms that penalize the Hessian of the loss function. Empirical results are provided to demonstrate the advantages of MPGD.

## 1 Introduction

Many important problems in supervised learning can be expressed by the following population risk minimization problem:

$$\min_{x \in \mathbb{R}^d} \Big\{ \mathcal{R}(x) := \mathbb{E}_{z \sim \mathcal{D}}[\ell(x, z)] \Big\}, \tag{1}$$

where $x \in \mathbb{R}^d$ denotes the learnable parameter, $z \in \mathcal{Z}$ denotes a data sample coming from an unknown data distribution $\mathcal{D}$, $\mathcal{Z}$ denotes the space of data points, and $\ell : \mathbb{R}^d \times \mathcal{Z} \to \mathbb{R}_+$ is the loss

36th Conference on Neural Information Processing Systems (NeurIPS 2022).

function, measuring the quality of the parameter $x$. Since the data distribution $\mathcal{D}$ is not known, as a proxy for solving (1), we consider the empirical risk minimization (ERM) problem, given as follows:

$$\min_{x \in \mathbb{R}^d}\Big\{\hat{\mathcal{R}}(x, S_n) := \frac{1}{n}\sum_{i=1}^{n}\ell(x, z_i)\Big\}, \tag{2}$$

where $S_n := \{z_1, \ldots, z_n\}$ denotes a *training set* of $n$ points that are independently and identically distributed (i.i.d.) and sampled from $\mathcal{D}$. To solve the ERM problem, typically gradient-based optimization algorithms are being used in practice.

In this study, we will consider the gradient descent (GD) algorithm [C+47], which is based on the following deterministic recursion:

$$x_{k+1} = x_k - \eta_k \nabla \hat{\mathcal{R}}(x_k, S_n), \quad k = 0, 1, 2, \ldots. \tag{3}$$

Here, $k$ represents the iteration number, $\eta_k > 0$ is the step-size (learning rate), and $x_0 \in \mathbb{R}^d$ is the initialization. Even though stochastic counterparts of GD (e.g., stochastic GD [BCN18]) have been more popular due to their reduced computational requirements, the GD algorithm has attracted increasing attention in the past few years, as it has been illustrated that GD can achieve similar generalization performance, as measured by the *generalization error* $|\hat{\mathcal{R}}(x_k, S_n) - \mathcal{R}(x_k)|$, as long as certain design choices are implemented [GGP+21]. These design choices mainly include using large and/or oscillating learning rates and using explicit regularization of the empirical risk.

Contrary to the wisdom of optimization theory, which typically suggests using smaller learning rates, the fact that large learning rates often provide better generalization performance in modern non-convex optimization problems has opened up several research directions. In this context, [CKL+21] showed that large learning rates drive the GD dynamics towards the 'edge of stability', meaning that the learning rate should be chosen large enough to make the algorithm diverge. Recently, the large learning rates have been extended to exotic learning rate schedules (i.e., non-trivial evolution of $\eta_k$), such as fractal scheduling [AGZ21] and cyclic scheduling [Oym21, Smi17].

It has been shown that the use of large learning rates introduces a *chaotic behavior* in the GD dynamics [KT20]. Hence, a possible explanation for the improvements brought by large learning rates can be partially attributed to such chaotic behavior, since chaotic systems exhibit a stochastic-like evolutions (even though they are fully deterministic), which might be an underlying cause of the improvements in generalization obtained in non-convex settings. However, achieving the desired chaotic dynamics is a problem-dependent task and can be often difficult in practice.

In this paper, we develop a novel optimization framework, coined *multiscale perturbed gradient descent* (MPGD), where a chaotic component is introduced to GD in a controlled manner. Our approach is based on extending the GD recursion via generic regularization terms, where the regularization coefficients are modulated by using external deterministic dynamical systems that exhibit chaotic characteristics. Our main contributions are as follows.

- We build up on recent advances in rough paths theory and homogenization techniques [GM13, CFK+16, CFKM20], and show that, under appropriate assumptions, as the step-size decreases, the MPGD recursion converges weakly to a stochastic differential equation (SDE) driven by a heavy-tailed Lévy-stable process (Theorem 4.1; see also Theorem C.1 in Appendix).

- We draw connections to the recent theoretical links between heavy-tailed processes and generalization [SSG19, ŞSDE20, HŞKM21], and under certain topological regularity assumptions, we derive a generalization bound (Theorem 4.2) for the limiting SDE and relate the worst-case generalization error over its trajectories to the parameters of MPGD.

- We show that MPGD exhibits a natural form of implicit regularization by deriving an appropriate explicit regularizer in the weak perturbation regime (Theorem 4.3). Our analysis shows that, in this regime, the derived regularizer introduces terms that penalize the Hessian of the loss function.

- We empirically demonstrate the advantages of MPGD on different settings and provide further support for the developed framework and the theory (see Section 5).

We emphasize that our primary goal is to understand the behavior of deterministic GD in a chaotic setting. In this respect, we shall underline that, at this stage, our goal is not to develop a competitive algorithm that would outperform SGD, but rather to provide a solid theoretical analysis for how stochastic behavior can emerge from GD with suitable deterministic components, as well as their implication in terms of generalization error.

## 2 Preliminaries and Technical Background

**Notation.** For a vector $v \in \mathbb{R}^d$, we denote its $i$th component by either $v^i$ or $[v]^i$ and the Euclidean norm by $|v|$ (or $\|v\|$). diag(v) denotes the diagonal matrix with $i$th diagonal entry of $v^i$. $\langle \cdot, \cdot \rangle$ denotes the dot product of two vectors. $\mathcal{U}(d)$ denotes uniform distribution on $d$ dimension, $tr$ denotes trace, and the superscript $^T$ denotes transposition. $\circ$ and $\odot$ denote composition and Hadamard product respectively. $\nabla$ and $\nabla^2$ denote gradient and Hessian respectively. $\mathbb{E}$ denotes expectation. $\overset{(d)}{=}$ and $\overset{(d)}{\to}$ denote equivalence and convergence in the sense of distribution respectively. $I$ denotes the identity matrix, and $1_A$ denotes indicator function of the set $A$.

**Stable distributions.** Let $\alpha \in (0, 2]$ be a stability parameter and $d \geq 1$. A random variable $X \in \mathbb{R}^d$ is $\alpha$-*stable* if $X_1, X_2, \ldots$ are independent copies of $X$, then $n^{-1/\alpha} \sum_{j=1}^n X_j \overset{(d)}{=} X$ for all $n \geq 1$ [Sam17]. Stable distributions appear as the limiting distribution in the generalized central limit theorem (CLT) [GK54]. The case $\alpha = 2$ corresponds to the Gaussian distribution, while $\alpha = 1$ corresponds to the Cauchy distribution. The $p$-th moment of a stable random variable is finite if and only if $p < \alpha$. Here we are interested in the case $\alpha \in (1, 2)$, such that $\mathbb{E}[|X|] < \infty$ and $\mathbb{E}[|X|^2] = \infty$. If $X$ is symmetric, then there exists a scale parameter $c > 0$ and the characteristic function of $X$ is given by $\mathbb{E}\left[\exp(i\langle \xi, X \rangle)\right] = \exp(-|c\xi|^\alpha)$ for all $\xi \in \mathbb{R}^d$. If $X$ is a random vector with independent components, then there exists a scale vector $\mathbf{c} = (c^1, \ldots, c^d) \in \mathbb{R}^d_+$ and the characteristic function of $X$ is given by $\mathbb{E}\left[\exp(i\langle \xi, X \rangle)\right] = \exp(-\sum_{j=1}^d |c^j \xi^j|^\alpha)$.

**Lévy processes.** Lévy processes are stochastic processes $(L_t)_{t \geq 0}$ with independent and stationary increments, and are defined as follows [App09]:

1. For $N \in \mathbb{N}$ and $t_0 < t_1 < \ldots < t_N$, the increments $(L_{t_i} - L_{t_{i-1}})$ are independent for all $i$.
2. For any $t > s > 0$, $(L_t - L_s)$ and $L_{t-s}$ have the same distribution.
3. $L_t$ is continuous in probability, i.e., for all $\delta > 0$ and $s \geq 0$, $\mathbb{P}(|L_t - L_s| > \delta) \to 0$ as $t \to s$.

Typical examples of Lévy processes include Brownian motion and the $\alpha$-stable processes. By the Lévy-Khintchine formula, a Lévy process $(L_t)_{t \geq 0}$ with $L_0 = 0$ is determined by a triplet $(b, \Sigma, \nu)$ for some $b \in \mathbb{R}^d$, $\Sigma \in \mathbb{R}^{d \times d}$ positive semi-definite and a measure $\nu$ on $\mathbb{R}^d \setminus \{0\}$ such that $\int_{x \neq 0} \min\{1, |x|^2\} \nu(dx) < \infty$. The characteristic function of $L_t$ is therefore $\exp(-t\Psi(\xi))$, with the characteristic exponent $\Psi : \mathbb{R}^d \to \mathbb{C}$ defined by

$$\Psi(\xi) = -i\langle b, \xi \rangle + \frac{1}{2}\langle \xi, \Sigma\xi \rangle + \int_{\mathbb{R}^d} \left[1 - e^{i\langle x, \xi \rangle} + i\langle x, \xi \rangle 1_{|x| < 1}\right] \nu(dx). \tag{4}$$

In the Lévy–Itô decomposition of a Lévy process, $b$ denotes a constant drift, $\Sigma$ is the covariance matrix of a Brownian motion and $\nu$ characterises a Lévy jump process, which is modified to be càdlàg (right-continuous and has left limits everywhere) with countably many jumps. In particular, if $b = 0_{\mathbb{R}^d}$, $\Sigma = 0_{\mathbb{R}^{d \times d}}$ and $\nu(dx) = |x|^{-\alpha-1}dx$ for some $\alpha \in (0, 2)$, then the triplet $(0_{\mathbb{R}^d}, 0_{\mathbb{R}^{d \times d}}, |x|^{-\alpha-1}dx)$ gives a symmetric $\alpha$-stable process, denoted by $(L_t^\alpha)_{t \geq 0}$.

**Marcus differential equations.** Passing to the limit of the driving processes with jumps in an SDE is not trivial, as one needs to be careful with the meaning of integration [CP14]. Unfortunately, the common Itô and Stratonovich integration (which are based on Riemann-Stieltjes sums with the integrand evaluated at the left end point and the midpoint of the partition intervals respectively) fail to provide the desired convergence. To this end, we resort to Marcus differential equations [Mar81], which in their basic form are given by:

$$\mathrm{d}X_t = b(X_t) \diamond \mathrm{d}L_t,$$

where $\diamond$ denotes integration in the Marcus sense. Marcus integrals involve sums over infinitely many jumps and transform under the usual laws of calculus [App09], and thus play a similar role for Lévy processes as the Stratonovich integral for Brownian motion. The solution map of Marcus differential equations is continuous with respect to the driving process under certain variants of the Skorokhod topology. Note that if $b(X_t)$ is constant in $X_t$, then there is no difference in the solutions of the SDE in the sense of Marcus, Itô and Stratonovich. We refer to Appendix B for more details on these.

**Generating stable laws using the Thaler map.** Let $\gamma := 1/\alpha$. It was shown in [GM21] that $\alpha$-stable laws with $\alpha \in (1, 2)$ can be generated using a deterministic dynamical system whose states

$y_k$ are obtained by iterating the Thaler map $T : [0, 1] \to [0, 1]$ [Tha80]:

$$T(y) = (y^{1-\gamma} + (1 + y)^{1-\gamma} - 1)^{1/(1-\gamma)} \mod 1 \,,\, y_{k+1} = T(y_k), \tag{5}$$

with any source of randomness coming solely from the initial condition $y_0 \in [0, 1]$. The map has the following properties. Let $y^* \in (0, 1)$ be the unique solution to the equation

$$(y^*)^{1-\gamma} + (1 + y^*)^{1-\gamma} = 2.$$

There are two increasing branches on the intervals $[0, y^*]$, $[y^*, 1]$. For $\gamma \in [0, 1)$, there exists a unique invariant probability density $h(y) = \frac{1-\gamma}{2^{1-\gamma}}(y^{-\gamma} + (1 + y)^{-\gamma})$. Moreover, the density defines a finite measure. For $\gamma \in (0, 1)$, the map is nonuniformly expanding with a neutral fixed point at $y = 0$ and correlations decay algebraically with rate $1/k^{\alpha-1}$ (which is sharp) [GM21]. Heuristically, the trajectory spends prolonged period of times near $y = 0$ (laminar dynamics near the fixed point), slowing down the decay of correlation as $\alpha$ decreases.

Let the observable $v : [0, 1] \to \mathbb{R}$ be a Hölder function with zero mean with respect to the invariant measure of $T$, i.e., $d\nu = h \, dy$, and consider the sequence of random variables $(v \circ T^k(y_0))_{k \geq 0}$, with the randomness coming from the initial state $y_0$. When $\gamma \in (1/2, 1)$, the correlations are not summable and CLT breaks down if $v(0) \neq 0$ (it "sees" the fixed point at $y = 0$). Heuristically, the Birkhoff sum $\sum_{j=0}^{k-1} v \circ T^j$ is ballistic with almost linear behavior near $x = 0$ and the small jumps of size $v(0)$ accumulate into a single large jump, contradicting CLT [GM21]. Instead, if $v$ is properly normalized, then it was shown in [Gou04] that the one-sided stable limit law $\frac{1}{k^\gamma} \sum_{j=0}^{k-1} v \circ T^j \overset{(d)}{\to} X_{\alpha,\beta}$ as $k \to \infty$, with $\beta = \text{sign}(v(0))$, holds instead, where $X_{\alpha,\beta}$ denotes the stable law whose characteristic function is given by:

$$\mathbb{E}[e^{itX_{\alpha,\beta}}] = \exp(-|t|^\alpha (1 - i\beta \, \text{sign}(t) \tan(\alpha\pi/2))). \tag{6}$$

In this paper, we focus on MPGD with chaotic components whose limiting behavior at each time is described by the class of stable laws $X_{\alpha,\beta}$ whose characteristic function is given by (6) with the stability parameter $\alpha \in (1, 2)$ (i.e., $\gamma \in (1/2, 1)$) and the skewness parameter $\beta \in [-1, 1]$. The $X_{\alpha,\beta}$ is centered (i.e., $\mathbb{E}X_{\alpha,\beta} = 0$), totally skewed (or one-sided) if $\beta = \pm 1$, and symmetric if $\beta = 0$.

## 3   Multiscale Perturbed Gradient Descent (MPGD)

In full generality, we start by considering the following family (parametrized by $m > 0$) of deterministic fast-slow dynamical systems on $\mathbb{R}^d \times Y$, with $Y$ a bounded metric space:

$$\begin{cases} x_{k+1}^{(m)} = x_k^{(m)} + \dfrac{1}{m} a_m(x_k^{(m)}) + \sum_{i=1}^{q} \dfrac{1}{m^{1/\alpha_i}} b_m^{(i)}(x_k^{(m)}) v_i(y_k^{(i)}), \\ y_{k+1}^{(i)} = T_i(y_k^{(i)}), \;\; i = 1, 2, \ldots, q, \end{cases} \tag{7}$$

for $k = 0, 1, 2, \ldots$, where the $\alpha_i \in (1, 2)$, $a_m : \mathbb{R}^d \to \mathbb{R}^d$, $b_m^{(i)} : \mathbb{R}^d \to \mathbb{R}^{d \times r_i}$, $v_i : Y \to \mathbb{R}^{r_i}$, and $T_i : Y \to Y$. The initial conditions $x_0^{(m)}, y_0^{(i)}$ are independent random variables. One important example of recursions of the form (7) is the GD defined in (3). Other than GD, the above family of recursions (7) includes various variants of GD as special cases upon specifying suitable choices of the $a_m, b_m^{(i)}, v_i, T_i$, and the dimensions.

Of particular interest are new algorithms that can be studied within the above framework. To this end, we introduce *multiscale perturbed gradient descent* (MPGD), whose dynamics are described by the following recursion:

$$\begin{cases} x_{k+1}^{(m)} = x_k^{(m)} - \dfrac{1}{m} \nabla \hat{\mathcal{R}}(x_k^{(m)}, S_n) - \dfrac{\mu}{m^{\frac{1}{\alpha_1}}} v_1(y_k^{(1)}) \odot x_k^{(m)} + \dfrac{\sigma}{m^{\frac{1}{\alpha_2}}} v_2(y_k^{(2)}), \\ y_{k+1}^{(1)} = T(y_k^{(1)}), \;\; y_{k+1}^{(2)} = T(y_k^{(2)}), \;\; k = 0, 1, 2, \ldots, \end{cases} \tag{8}$$

where $T$ is the Thaler map (5) with $\gamma = \alpha^{-1} \in (1/2, 1)$, the $v_1, v_2$ are observable maps, $\mu \geq 0$ and $\sigma \in \mathbb{R}$ are tunable parameters. The above recursion is of the form (7), with $q = 2$, $r_1 = r_2 = d$,

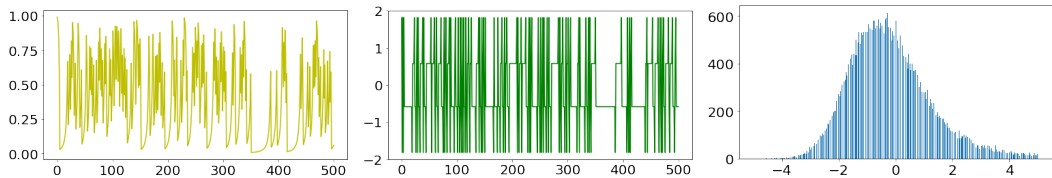

Figure 1: A realization of the Thaler iterates (left) and the observable $v$ (middle) for $\gamma = 0.6$, $\beta = 0.1$. Empirical distribution of the corresponding Birkhoff sum $\frac{1}{k^\gamma} \sum_{j=0}^{k-1} v^{(j)}$ with $k = 1000$ (right).

$a_m(x) = -\nabla \hat{\mathcal{R}}(x, S_n)$, $b_m^{(1)}(x) = -\mu \operatorname{diag}(\mathrm{x})$, and $b_m^{(2)}(x) = \sigma I$ independent of $m$. Another class of algorithms that falls within this framework is when $m$ is set to $n$, the number of training data points, in which case the empirical risk converges to the population risk as $m = n \to \infty$.

To complete the description of MPGD, it remains to specify the choice for the observables $v_i(y_k^{(i)})$ in (8) whose rescaled version can be shown to generate stable laws $X_{\alpha,\beta}$, with $\alpha \in (1, 2)$ and $\beta \in [-1, 1]$, in the limit $m \to \infty$. Following [GM21], we take them to be independent realizations of the observable $v^{(k)}$ described as follows. Let $d_\alpha = \frac{\alpha^\alpha (1-\gamma) \Gamma(1-\alpha) \cos(\alpha\pi/2)}{2^{1-\gamma}-1}$, and $\delta_0, \delta_1, \dots$ be independent copies of the random variable $\delta$ where $\mathbb{P}(\delta = \pm 1) = \frac{1}{2}(1 \pm \beta)$. The observables $v^{(k)}$ are then defined to be: $v^{(k)} = \chi^{(k)} v \circ T^k$, where $v : [0, 1] \to \mathbb{R}$ is the mean zero observable given by $v(y) = d_\alpha^{-\gamma}(1 - 2^{\gamma-1})^{-\gamma} \tilde{v}(y)$, with

$$\tilde{v}(y) = \begin{cases} 1 & \text{if } y \leq y^*, \\ (1 - 2^{1-\gamma})^{-1} & \text{if } y > y^*, \end{cases} \quad \chi^{(k)} = \chi_{k-1} \cdots \chi_0 \in \{\pm 1\} \text{ with } \chi_j = \begin{cases} 1 & \text{if } T^j y \leq y^*, \\ \delta_j & \text{if } T^j y > y^*. \end{cases}$$

In particular, the random variables $\chi^{(k)}$ get updated only when the trajectory visits $(y^*, 1]$ and are not changed during the phase in $[0, y^*]$.

The initial condition can be, in theory, equally well chosen using the invariant probability measure $\nu$ or the uniform Lebesgue measure. Empirically, convergence of the probability density is faster if it is drawn using $\nu$. Hence, we consider initial conditions drawn using $\nu$. Therefore, for MPGD we propagate uniformly distributed initial conditions $y_0' \in [0, 1]$ under 10,000 iterations of the Thaler map and take the initial condition as $y_0 = T^{10000} y_0'$. Figure 1 illustrates the dynamics of a realization of the Thaler iterates and the observable (perturbations used in MPGD), as well as the asymmetry and heavy tail nature of the distribution of the corresponding Birkhoff sum.

## 4 Theoretical Analysis

In this section, we analyze MPGD via three different angles. We start by presenting results on superdiffusive limit (homogenization [PS08]) for a class of fast-slow deterministic systems that include MPGD as a special case in Subsection 4.1. We then study generalization properties of MPGD by deriving a generalization bound for the limiting dynamics in Subsection 4.2. Lastly, we study the regularizing effects of MPGD via the lens of implicit regularization in Subsection 4.3.

### 4.1 Superdiffusive Limits for the Fast-Slow Systems and MPGD

Without loss of generality, we set $q = 1$ in (7) and consider the following family of recursions:

$$\begin{cases} x_{k+1}^{(m)} = x_k^{(m)} + m^{-1} a_m(x_k^{(m)}) + m^{-\frac{1}{\alpha}} b_m(x_k^{(m)}) v(y_k), \\ y_{k+1} = T(y_k), \end{cases} \tag{9}$$

where $T$ is the Thaler map and $v$ is the observable constructed in Section 3.

The following result shows that stochastic dynamics emerges in the above family of deterministic dynamical systems in the limit $m \to \infty$. Moreover, the limiting dynamics are described by a stochastic differential equation (SDE) driven by a heavy-tailed Lévy-stable process.

**Theorem 4.1** (Superdiffusive Limit – Informal). *Let $\alpha \in (1, 2)$ and assume that the coefficients $a_m$, $b_m$ and the initial condition $x_0^{(m)}$ of (9) are well-behaved as $m \to \infty$. Then, under certain regularity*

*conditions on the $a_m$, $b_m$, the process $(x^{(m)}_{\lfloor mt \rfloor})_{t \geq 0}$ converges in distribution to the solution $(X_t)_{t \geq 0}$ of the SDE:*

$$\mathrm{d}X_t = a(X_t)\mathrm{d}t + b(X_t) \diamond \mathrm{dL}_t^{\alpha,\beta} \text{ with } X_0 = x_0 \in \mathbb{R}^d,$$

*as $m \to \infty$, where $a = \lim_{m\to\infty} a_m$, $b = \lim_{m\to\infty} b_m$, $x_0 = \lim_{m\to\infty} x_0^{(m)}$, $\diamond$ denotes the Marcus integration, defined in Section B.2.1, and $(\mathrm{L}_t^{\alpha,\beta})_{t \geq 0}$ is a Lévy process with the characteristic exponent*

$$\Psi(\xi) = |\xi|^\alpha (1 - i\beta \,\mathrm{sign}(\xi) \tan(\alpha\pi/2)). \tag{10}$$

In particular, Theorem 4.1 implies convergence in distribution of $x_{\lfloor mt \rfloor}^{(m)}$ to $X_t$ at fixed $t$. It is a special case of Theorem C.1, which covers a large class of the maps $T$ and observables $v$, and extends the results in [GM21, CFKM20] to the case where the coefficients are dependent on $m$. We provide technical details on the topology used for the convergence and the proof in Appendix B-C.

Specializing Theorem 4.1 (extended to the case of $q = 2$) to the MPGD recursion (8), we see that the rescaled MPGD iterate process, $(x_{\lfloor mt \rfloor}^{(m)})_{t \geq 0}$, converges in distribution to the solution $(X_t)_{t \geq 0}$ of the following SDE:

$$\mathrm{d}X_t = -\nabla\hat{\mathcal{R}}(X_t, S_n)\mathrm{d}t - \mu \, \mathrm{diag}(X_t) \diamond \mathrm{dL}_t^{\alpha_1,\beta_1} + \sigma \mathrm{d}\tilde{\mathrm{L}}_t^{\alpha_2,\beta_2}, \quad X_0 = \lim_{m\to\infty} x_0^{(m)},$$

in the limit $m \to \infty$, which is equivalent to sending the step-size, $\eta := 1/m$, in the MPGD to zero. Here $(\mathrm{L}_t^{\alpha_1,\beta_1})_{t \geq 0}$ and $(\tilde{\mathrm{L}}_t^{\alpha_2,\beta_2})_{t \geq 0}$ are independent Lévy processes with characteristic exponent (10), and $\beta_1$, $\beta_2 \in [-1, 1]$ are the parameters appearing in the construction of the observables $v_1$ and $v_2$ respectively. Therefore, we see that MPGD converges to a gradient flow driven by heavy-tailed Lévy-stable processes in the considered scaling limit. The driving processes contain both additive and multiplicative components. The multiplicative component can be interpreted as a dynamical version of the weight decay, whereas the additive component can be viewed as gradient noise [NVL$^+$15].

## 4.2 Generalization Properties of MPGD

In this section, we shift our focus to the generalization properties of MPGD. Our main roadmap will be to consider the limiting SDE arising from Theorem 4.1 (with $\beta = 0$) and relate it to the recently developed generalization bounds for heavy-tailed random processes [ŞSDE20, HŞKM21].

For mathematical convenience, we consider the following special case of (7)[1]:

$$\begin{cases} x_{k+1}^{(m)} = x_k^{(m)} + m^{-1}a_m(x_k^{(m)}) + m^{-\frac{1}{\alpha}}b_m(x_k^{(m)}) \left(v(y_k^{(1)}) - v(y_k^{(2)})\right), \\ y_{k+1}^{(1)} = T(y_k^{(1)}), \quad y_{k+1}^{(2)} = T(y_k^{(2)}) \end{cases} \tag{11}$$

In the light of Theorem 4.1, the recursion (11) converges weakly to the following SDE:

$$\mathrm{d}X_t = a(X_t)\mathrm{d}t + b(X_t) \diamond \mathrm{dL}_t^{\alpha}, \tag{12}$$

where $\mathrm{L}_t^\alpha$ denotes the *symmetric $\alpha$-stable process* in $\mathbb{R}^d$ (i.e., $\beta = 0$).

Following the previous work [ŞSDE20, HŞKM21], we are interested in bounding the worst-case generalization error over the trajectories of (12). More precisely, let $(X_t)_{0 \leq t \leq 1}$ denote the solution of (12)[2], and define the set $\mathcal{X}$ so that it contains all the points visited by the trajectory:

$$\mathcal{X} := \{x \in \mathbb{R}^d : \exists t \in [0,1], X_t = x\}. \tag{13}$$

Then, our goal is to analyze $\sup_{x \in \mathcal{X}} |\hat{\mathcal{R}}(x, S_n) - \mathcal{R}(x)|$. Since we are mainly interested in the case when $a(x) = -\nabla\hat{\mathcal{R}}(x, S_n)$, the SDE (12) and hence the trajectory $\mathcal{X}$ will depend on the dataset $S_n$. Therefore, in the generalization bound, we will need to control the statistical dependence of $\mathcal{X}$ on $S_n$, through the notion of $\rho$-mutual information, defined as follows. Let $X, Y$ be two random elements, let $\mathbb{P}_{X,Y}$ denote their joint distribution, and let $\mathbb{P}_X, \mathbb{P}_Y$ denote their respective marginal distributions. Then the $\rho$-mutual information between $X$ and $Y$ is defined as: $I_\rho(X, Y) = D_\rho(\mathbb{P}_{X,Y} \| \mathbb{P}_X \otimes \mathbb{P}_Y)$,

---

[1]We believe a similar result would hold for (9) without symmetrization, which we leave for future work.

[2]Here, the range $[0, 1]$ is arbitrary and could be replaced with any range $[0, T]$ for $0 < T < \infty$.

where $D_\rho$ is the $\rho$-Renyi divergence:[3] $D_\rho(\mu, \nu) = \frac{1}{\rho-1} \log \mathbb{E} \left[ \frac{d\mu}{d\lambda}(Z)^\rho \frac{d\nu}{d\lambda}(Z)^{1-\rho} \right]$, and $\frac{d\mu}{d\lambda}$ denotes the Radon-Nikodym derivative.

Finally, we require two technical regularity assumptions (Assumptions 2 and 3, given in Appendix D) for our generalization bound. Informally, Assumption 2 is a topological regularity condition over the trajectory $\mathcal{X}$, whereas Assumption 3 is a statistical regularity assumption and imposes that in the neighborhood of a local minimum $x^\star$ of $\hat{\mathcal{R}}(x, S_n)$, the statistical behavior of the process $X_t$ can be approximated by the process solving the SDE $d\tilde{X}_t = b(x^\star) \diamond dL_t^\alpha$. This requires $b$ to be regular in the neighborhood of $x^\star$ and further imposes that $a(x^\star) = 0$, which is natural when $a(x) = -\nabla \hat{\mathcal{R}}(x, S_n)$. We now ready to present our generalization bound.

**Theorem 4.2** (Generalization bound). *Assume that Assumptions 2 and 3 given in Section D hold, and that $\ell$ is bounded by $B > 0$ and $L$-Lipschitz continuous. There exists a constant $K_1 > 0$ such that with probability at least $1 - \delta$,*

$$\sup_{x \in \mathcal{X}} |\hat{\mathcal{R}}(x, S_n) - \mathcal{R}(x)| \le K_1 \max\{B, L\} \left( \sqrt{\frac{\alpha}{n}} + \sqrt{\frac{\log(1/\delta) + I_\infty(S_n, \mathcal{X})}{n}} \right), \quad (14)$$

*where $I_\infty(X, Y) := \lim_{\rho \to \infty} I_\rho(X, Y)$. Furthermore, there exists $K_2 > 0$ such that*

$$\mathbb{E} \sup_{x \in \mathcal{X}} |\hat{\mathcal{R}}(x, S_n) - \mathcal{R}(x)| \le K_2 \max\{B, L\} \left( \sqrt{\frac{\alpha}{n}} + \sqrt{\frac{I_1(S_n, \mathcal{X})}{n}} \right). \quad (15)$$

This result shows that the generalization error is mainly bounded by two terms: the tail exponent of the limiting Lévy process $L_t^\alpha$, and the statistical dependence of $\mathcal{X}$ on $S_n$. The bound suggests that, provided the mutual information between $\mathcal{X}$ and $S_n$ is fixed, the generalization error can be reduced by using a smaller $\alpha$ in MPGD. However, it has been illustrated that a smaller $\alpha$ can induce significant error on $\hat{\mathcal{R}}(x, S_n)$ [SZTG20, CWZ+21], hence a reasonable value of $\alpha$ should be chosen to balance both the empirical risk and the generalization error. The algorithm parameters $\mu, \sigma$ interact with the bound via the constant terms $K_1, K_2$ and the mutual information terms $I_1, I_\infty$. Unfortunately, their dependence is quite implicit and we are not able to provide quantitative estimations.

Finally, the main difference between Theorem 4.2 and [ŞSDE20, HŞKM21] is that, in our bounds we have explicit access to the characteristic exponent of $L_t^\alpha$ since we construct it manually through our dynamics; whereas in the prior work generic processes were used as approximations for SGD, hence their results are not as explicit.

### 4.3 Implicit Regularization of MPGD

To understand MPGD and its connection to the vanilla GD (3) better, we provide additional analysis for the behavior of loss functions optimized under the recursion (8) via the lens of implicit regularization [Mah12, LEHM21]. By this, we mean regularization imposed implicitly by the learning strategy, without explicitly modifying the loss. We shall achieve this by deriving an appropriate explicit regularizer through a perturbation analysis in the weak perturbation regime.

To this end, we work in the regime where the parameters $\mu, \sigma$ are small, keeping $m$ fixed. We set $\mu = \mu_0 \epsilon$ and $\sigma = \sigma_0 \epsilon$, where $\epsilon > 0$ is a small parameter, and analyze the loss optimized using the recursion (8) in the small $\epsilon$ regime. In the sequel, we let $\overline{x}_k^{(m)}$ denote the states of the unperturbed GD, satisfying the recursion $\overline{x}_{k+1}^{(m)} = \overline{x}_k^{(m)} - \frac{1}{m} \hat{\mathcal{R}}(\overline{x}_k^{(m)}, S_n)$, for $k = 0, 1, \ldots$. To simplify notation, we shall denote $\hat{\mathcal{R}}(x) := \hat{\mathcal{R}}(x, S_n)$.

The following result relates the loss function, averaged over realizations of the injected perturbations, evolved under training with MPGD to that of unperturbed GD in the weak perturbation regime.

**Theorem 4.3** (Implicit regularization). *Let $m > 0$, $k \in \mathbb{N}$, and $S_n$ be given. Assume that $r_1 = 1$ and $r_2 = d$ in Eq. (8), and $\mathbb{E}[v_1(y_k^{(1)})] = \mathbb{E}[v_2(y_k^{(2)})] = 0$ for all $k$. Then, for a scalar-valued loss function $\hat{\mathcal{R}}$ with $\nabla \hat{\mathcal{R}}$ having Lipschitz continuous partial derivatives in each coordinate up to order*

---

[3]As $\rho \to 1$, $D_\rho$ tends to the Kullback-Leibler divergence.

*three (inclusive),*

$$\mathbb{E}[\hat{\mathcal{R}}(x_k^{(m)})] = \hat{\mathcal{R}}(\overline{x}_k^{(m)}) + \frac{\epsilon^2}{2}\left(tr\left(C_k^{(m)}\nabla^2\hat{\mathcal{R}}(\overline{x}_k^{(m)})\right) - \nabla\hat{\mathcal{R}}(\overline{x}_k^{(m)})^T\lambda_k^{(m)}\right) + \mathcal{O}(\epsilon^3), \quad (16)$$

*as $\epsilon \to 0$, where the expectation is with respect to the realization of the $y^{(i)}$, the $\lambda_k^{(m)}$ are the vector with the lth component:*

$$[\lambda_k^{(m)}]^l = \frac{1}{m}\sum_{i=1}^{k}\sum_{j=1}^{d}[\Phi_i^{(m)}]^{lj}tr\left(C_{i-1}^{(m)}\nabla^2[\nabla\hat{\mathcal{R}}(\overline{x}_{i-1}^{(m)})]^j\right),$$

*the $\Phi_i^{(m)} := \prod_{j=i}^{k-1}(I - \frac{1}{m}\nabla^2\hat{\mathcal{R}}(\overline{x}_j^{(m)}))$, with the empty product taken to be the identity, and the $C_k^{(m)}$ are covariance matrices with the $(p,q)$-entry of $\sum_{i_1,i_2=1}^{k}\sum_{r,s=1}^{d}[\Phi_{i_1}^{(m)}]^{pr}[\Phi_{i_2}^{(m)}]^{qs}\theta_{i_1,i_2,r,s}^{(m)}$, where*

$$\theta_{i_1,i_2,r,s}^{(m)} = \frac{\mu_0^2}{m^{2/\alpha_1}}\mathbb{E}[v_1(y_{i_1-1}^{(1)})v_1(y_{i_2-1}^{(1)})]\cdot[x_{i_1-1}^{(m)}]^r[x_{i_2-1}^{(m)}]^s + \frac{\sigma_0^2}{m^{2/\alpha_2}}\mathbb{E}[[v_2]^r(y_{i_1-1}^{(2)})[v_2]^s(y_{i_2-1}^{(2)})].$$

Theorem 4.3 implies that the loss function optimized under MPGD at a given iteration and $m$ is, on average, approximately equivalent to a regularized objective functional. Moreover, $\frac{\nabla\hat{\mathcal{R}}(\overline{x}_k^{(m)})^T v_k^{(m)}}{tr\left(C_k^{(m)}\nabla^2\hat{\mathcal{R}}(\overline{x}_k^{(m)})\right)} \le \frac{C}{m}$ for some constant $C$ independent of $m$, suggesting that the trace term is the dominant explicit regularizer for large $m$. This explicit regularizer is solely determined by the discrete-time flow generated by the Jacobians $I - \frac{1}{m}\nabla^2\hat{\mathcal{R}}(\overline{x}_j^{(m)})$, the covariances $C_k^{(m)}$, and the Hessian of the loss function, all evaluated along the dynamics of the unperturbed GD. We can therefore expect the use of the perturbations in MPGD as a regularization mechanism should reduce the Hessian of the loss function according to the perturbation levels $(\mu_0, \sigma_0)$ and the correlations $\frac{1}{m^{2/\alpha_1}}\mathbb{E}[v_1(y_{i_1-1}^{(1)})v_1(y_{i_2-1}^{(1)})]$, $\frac{1}{m^{2/\alpha_2}}\mathbb{E}[v_2(y_{i_1-1}^{(2)})v_2(y_{i_2-1}^{(2)})]$. Reducing the Hessian of the loss function can lead to flatter minima in the loss landscape, which is widely believed to associate with better generalization in deep learning [KMN+16, JNM+19].

Lastly, we observe that in the weak perturbation regime and for non-convex losses, MPGD is more beneficial than GD in the sense that it reduces the Hessian of the loss, thereby promoting flatter minima. Hence, in order to observe the benefits of our scheme, we suspect that the optimization problem should be non-convex and local minima should have different curvatures.

## 5 Empirical Results

In this section, we illustrate the advantages of MPGD compared to other schemes on the tasks of (1) minimizing the widening valley loss, (2) regression on the Airfoil Self-Noise Dataset, and (3) classification on CIFAR-10. We also provide additional results and details in Appendix F.

### 5.1 Minimizing the Widening Valley Loss

We consider the problem of minimizing the widening valley loss, given by $\ell(u,v) = v^2\|u\|^2/2$. The gradient of $l$ is given by $\nabla\ell(u,v) = (v^2 u, \|u\|^2 v)$, and the trace of the Hessian is $dv^2 + \|u\|^2$, where $d$ is the dimension of $u$. The trace of the Hessian measures the flatness of the minima, which is monotonously changing in this case. The loss has a valley of minima with zero loss for all $(u,v)$ with $v = 0$. The smaller the norm of $u$, the flatter the minimum. Vanilla GD gets stuck when it first enters this valley, and the intuition is that injecting suitable perturbations should help convergence to a flat (with small $\|u\|$) part of the valley. All $(u,v)$ with $v = 0$ are minima, but we also need $\|u\|$ to be minimized in order to minimize the trace of the Hessian. In high dimension (large $d$), the GD path is biased towards making $v$ small and not optimizing $u$ since the direction along $v$ is the most curved.

For the experiments, we start optimizing from the point $(u_0, 0)$, where $u_0 \sim 5 \cdot \mathcal{U}(d)$ with $d = 10$, and use the learning rate $\eta = 0.01$. We study and compare the behavior of the following schemes: (i) baseline (vanilla GD), (ii) GD with uncorrelated Gaussian noise injection instead, and (iii) MPGD. Figure 2 demonstrates that MPGD can lead to successful optimization of the widening loss whereas the baseline GD and GD with Gaussian perturbations lead to poor solutions. This is in agreement with our analysis of implicit regularization for MPGD, showing that the injected perturbations effectively favor small trace of the loss Hessian, thereby biasing the solution to flatter region of the loss landscape.

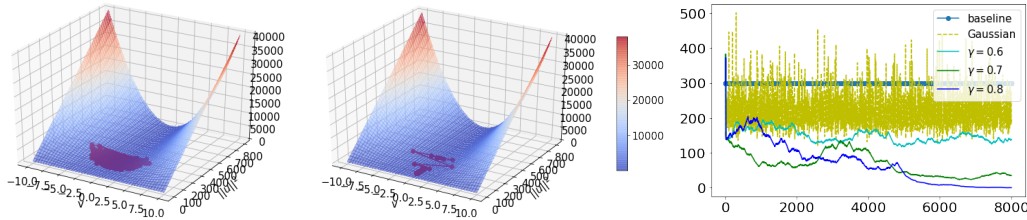

Figure 2: Evolution of GD with Gaussian perturbations (left plot) vs. that of MPGD with $\gamma = 0.7$, $\beta = 0.5$ (middle plot), using $\mu = 0.02$, $\sigma = 0.05$, and $\eta = 0.01$ ($m = 100$). Here we see that MPGD leads to successful optimization of the widening valley loss whereas that with Gaussian perturbations fails to converge. Moreover, MPGD effectively reduces the trace of loss Hessian (see right plot), steering the GD iterates to flatter region of the loss landscape.

## 5.2 Airfoil Self-Noise Prediction

We consider the Airfoil Self-Noise Dataset [DG17] from the UCI repository. It comprises of differently sized airfoils at various wind tunnel speeds and angles of attack. This is a regression problem aiming to predict scaled sound pressure level, in decibels, of the airfoil based on the the features: frequency, angle of attack, chord length, free-stream velocity, and suction side displacement thickness. We use 1202 samples for training and 301 samples for testing.

Table 1: Shallow neural nets trained on the airfoil data set. The results in parenthesis are achieved with the variant (11). All the results are averaged over 5 models trained with different seed values.

| Scheme | test RMSE | RMSE gap |
|---|---|---|
| Baseline | 0.4309 | 0.2411 |
| Gaussian | 0.4279 | 0.2354 |
| MPGD, $\gamma = 0.55$ | 0.3916 (0.3865) | 0.2256 (0.2305) |
| MPGD, $\gamma = 0.6$ | **0.3810** (0.4092) | 0.2298 (**0.2206**) |
| MPGD, $\gamma = 0.65$ | 0.3829 (0.3891) | 0.2407 (0.2307) |
| MPGD, $\gamma = 0.7$ | 0.4600 (0.3754) | 0.2315 (0.2311) |

For the training, we use a fully connected shallow neural network of width 16 with ReLU activation and train for 3000 epochs with the learning rate $\eta = 0.1$, using mean square error (MSE) as the loss and choosing $\beta = 0.5$. Table 1 reports the average root MSE (RMSE) and the RMSE gap (defined as test RMSE - train RMSE) evaluated for models that are trained with 5 different seed values for this task. We can see that MPGD leads to both lower test RMSE and RMSE gap when compared with vanilla GD (baseline) and GD with uncorrelated Gaussian perturbations (see the results not in parenthesis in Table 1; here $\mu = 0.01$, $\sigma = 0.02$). Using the form of the perturbations in (11) instead can also give lower RMSE gap (see the results in parenthesis in Table 1; here $\sigma = \mu = 0.01$). Overall these results support our generalization theory for MPGD. Lastly, we note that using larger values of $\gamma$ naively does not guarantee better test performance: one has to fine tune the parameters $\beta, \mu, \sigma, \eta$ appropriately to achieve favorable trade-off between training stability and test performance.

## 5.3 CIFAR-10 Classification

We consider training a ResNet-18 classifier [HZRS16] on the CIFAR-10 data set [Kri09]. We follow the setup used in [GGP+21], where a ResNet-18 is trained using batch size of 50K (the entire training dataset). As in [GGP+21], we consider a standard ResNet-18 with the parameters in the linear layer randomly initialized and the parameters in the batch normalization initialized with mean zero and unit variance, except for the last in each residual branch which is initialized to zero. The default random CIFAR-10 data ordering is kept as is, and every image is randomly augmented by horizontal flips and random crops after padding by 4 pixels.

Using the setup of [GGP+21], the reference mini-batch SGD is trained using a batch size of 128 (sampling without replacement), Nesterov momentum of 0.9 and weight decay of 0.0005. The learning rate is warmed up from 0.0 to 0.1 over the first 5 epochs and then reduced via cosine annealing to 0 over the course of training for 300 epochs (resulting in 390x300=117,000 update steps). The validation accuracy obtained in [GGP+21] is 95.7%. For the full-batch GDs, we replace the mini-batch updates by full batches and accumulate the gradients over all mini-batches. For the Gaussian scheme and MPGD, we inject the perturbations into the GD recursion for parameters of the input layer of ResNet-18, choosing $\mu = 0.03$, $\sigma = 0.01$ and $\beta = 0.5$ (for MPGD). In our experiments,

Table 2: ResNet-18 trained on CIFAR-10 for 1000 epochs. Here, accuracy gap = training accuracy - validation accuracy. The results in parenthesis are achieved with the variant of MPGD (11). All the results are averaged over 5 models trained with different seed values.

| Scheme | val. accuracy in % | accuracy gap in % |
|---|---|---|
| Baseline full batch GD | 73.10 | 8.99 |
| Gaussian | 70.58 | 8.53 |
| MPGD, $\gamma = 0.55$ | **75.97** (74.51) | 6.08 (6.58) |
| MPGD, $\gamma = 0.6$ | 75.78 (74.60) | **5.64** (6.15) |
| MPGD, $\gamma = 0.65$ | 74.39 (72.96) | 6.41 (6.85) |
| MPGD, $\gamma = 0.7$ | 73.72 (73.27) | 6.69 (6.04) |

we fix the learning rate to 0.1 throughout training, and do not add momentum and weight decay. This is in constrast to [GGP+21], where the additional tricks applied in the mini-batch SGD are also used for the full-batch GD, leading to higher test accuracies (see Table 2 in [GGP+21]).

Table 2 shows that MPGD can lead to better test performance and lower accuracy gap when compared to the baseline (full batch GD) and Gaussian noise-perturbed GD. Using the perturbations in (11) instead can also achieve lower accuracy gap. Therefore, adding perturbations of MPGD can be potentially a useful trick to improve training of deep architectures on benchmark data sets. Lastly, we note that additional tricks, such as gradient clipping and tuning of optimization hyperparameters, can be applied to improve our test accuracies and close the gap to results obtained with the reference mini-batch SGD. However, our main focus here is not on competitive performance, but rather on demonstrating the effects of the perturbations in MPGD when compared to full batch GDs and Gaussian noise-perturbed full-batch GD to support our theory.

Lastly, we remark that we did not tune the step-size for any of the algorithms. Since our generalization bound applies to the asymptotic SDE, which is obtained when the step-size goes to zero, in order to stay close to the theory, we chose a small enough step-size to be both not far from the continuous dynamics, and large-enough so that the algorithm converges in a reasonable amount of time. That being said, we have tried a range of step-sizes for both algorithms, and we observed that the proposed scheme consistently outperforms vanilla GD for smaller step-sizes. Whereas if we use a large step-size, both algorithms perform similarly. In this regime, vanilla GD as well potentially emits a chaotic behavior, which might also indicate the importance of the "implicit randomness". However, as we indicated in the introduction, this regime is not easily controllable, and our purpose is to introduce a controlled chaotic component, with a clear theoretical understanding.

## 6 Conclusion

In this paper, we introduce and study a class of slow-fast deterministic dynamical systems which homogenize to a limiting SDE driven by a heavy-tailed Lévy-stable process in an appropriate scaling limit as a rigorous framework for perturbed GDs. Within this framework, we introduce MPGD, a novel version of perturbed GD, which we show to have good generalization and regularization properties. We further demonstrate the advantages of MPGD empirically in various optimization tasks. Our framework can provide useful tools for identifying implicit randomness in deterministic optimization algorithms and inspire other promising algorithms. It would be interesting to investigate the interactions of the chaotic perturbations in MPGD with the noise arising from mini-batch sampling in SGD [WHX+20] and other optimization tricks. Since this is a theoretical paper studying a framework for perturbed GDs, there are no potential negative societal impacts of our work.

**Acknowledgements.** We are grateful to the computational resources provided by the Swedish National Infrastructure for Computing (SNIC) at Chalmers Centre for Computational Science and Engineering (C3SE) partially funded by the Swedish Research Council through grant agreement no. 2018-05973. S. H. Lim would like to acknowledge the WINQ Fellowship, the SNIC AI/ML grant, and the Swedish Research Council (VR/2021-03648) for providing support of this work. U. Ş.'s research was funded in part by the French government under management of Agence Nationale de la Recherche as part of the "Investissements d'avenir" program, reference ANR-19-P3IA-0001 (PRAIRIE 3IA Institute), and the European Research Council Starting Grant DYNASTY – 101039676.

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
