# OpenReview forum: "Chaotic Regularization and Heavy-Tailed Limits for Deterministic Gradient Descent"
_NeurIPS.cc/2022/Conference — NeurIPS 2022 Accept_

### Official Review · Reviewer_bLX6 · 2022-07-10

**Rating:** 6
**Confidence:** 3
**Soundness:** 3 good
**Presentation:** 3 good
**Contribution:** 3 good

**Summary:**

The paper considers a family of deterministic fast-slow dynamical systems. This family includes GD and some of its variants and the MPGD algorithm the authors introduce. The authors show that, as the learning rate vanishes, the MDGD dynamics converge weakly to an SDE driven by a heavy-tailed Levy stable process. Then, the authors derive a generalization bound for this limiting SDE for the symmetrical case. Finally, the authors analyze the implicit bias induced by these dynamics and show that, in a weak perturbation regime, it penalizes the Hessian of the loss function. These theoretical results are demonstrated empirically.

**Questions:**

Following my remark in the previous section, can you please elaborate on the different assumptions needed for the different theorems?

**Limitations:**

The authors mentioned some of the limitations of their results.

**Strengths And Weaknesses:**

*Strengths:*
The paper is clear and well written, and it establishes several interesting results, detailed above. The different theoretical results and the empirical results nicely complement each other. In addition, I believe that the framework the authors suggested might be used to analyze other optimization methods.

*Weaknesses:*
A lot of the main assumptions for the different Theorems only appear in the appendix and without justification or explanation. While I understand this is probably due to space constraints, this makes it hard to evaluate how general or reasonable the assumptions are and the limitations of the results.

---

> ### Author Response · Authors · 2022-08-02
> **Thank you for the careful review**
>
> We thank the reviewer for careful reading of the paper. As the reviewer correctly mentioned, we could not place the assumptions in the main text due to space constraints. However, in case our submission gets accepted, we will add the following discussion in the paper.
>
> 1. Regarding Assumption 1 for Theorem 4.1:
> - It is required that the difference between the gradients of coefficients $\\|\nabla (a_m- a)\\|_\infty$ goes to $0$ and also their H\"older constant $\frac{|\nabla (a_m-a)(x)- \nabla (a_m-a)(y)|}{|x-y|^\epsilon}\to 0$ uniformly on $x,y\in\mathbb{R}^m$ for some $\epsilon>0$. This translates to the assumptions that the loss is second differentiable and its Hessian is $\epsilon$-H\"older, which are reasonable assumptions.
> - Similarly, for the coefficients appearing before the $\alpha$-stable process (for $\alpha \in (1,2)$), it is required that
> $\\|\nabla (b_m- b)\\|_\infty \to 0$
> and
>
> $\sup_{x,y\in\mathbb{R}^m}\frac{|\nabla (b_m-b)(x)- \nabla (b_m-b)(y)|}{|x-y|^{\epsilon'}} \to 0$
>
> for some $\epsilon'>\alpha-1$. In the setup for MPGD (see Eq. (8) in the main paper) where $a_m = - \nabla \hat{\mathcal{R}}(x, S_n) = a$ (independent of $m$) and the $b_m$ is $-\mu \  diag(x)$ or $\sigma I$, it is easy to see that these assumptions are satisfied.
> - In terms of how general the assumption is, we note that Assumption 1 can cover more general situations; e.g., the case where
> $a_m$ is an empirical loss (taking $m = n$, the number of training samples) and $a$ is the  population loss, and therefore our framework can be adapted for analysis in other settings as well.
>
> 2. Regarding the assumptions for Theorem 4.2:
> - Assumption 2 is a geometric regularity condition over the trajectory of the multiscale perturbed gradient flow, which is common for random fractal processes given as solutions to stochastic differential equations; see [Hodgkinson et al. (2022)](https://arxiv.org/abs/2108.00781). This assumption ensures that the box-counting (Minkowski) dimension coincides with the Hausdorff dimension of the trajectory.
> - As for Assumption 3, it is natural to assume the existence of $x^*$ such that $a(x^*) = 0$ (since $a =  - \nabla \hat{\mathcal{R}}(x, S_n)$ in our MPGD setup). For the first point,  the positive definiteness of $b(x^*)$ can be satisfied by simply choosing $b$ to be the identity map. For the second point, recall that $\tilde{q}(x,\xi)=q(x,\xi)-\psi_{x^*}(\xi)$ and we require that:
>
> $$|\partial^\mathbf{j}_x \tilde{q}(x,\xi)|\le \Phi_\mathbf{j}(x)(1+\kappa_0|\xi|^\alpha) \text{ with }\mathbf{j}\in \mathbb{N}_0^m,\, |\mathbf{j}|\le m+1 \text{ for some }\Phi_\mathbf{j} \in L^1(\mathbb{R}^m); $$
>
> $X^*$ is an $\alpha$-stable L\'evy process, and its characteristic function $\psi_{X^*}(\xi)$ is given by $|\xi^T b(x^*)|^\alpha$ following Eq. (4) in the main text. Moreover, $q(x,\xi)$ is nothing but (4) with the $b, \Sigma, \mu$ depending on $x$. It is not hard to see that the above assumption holds if the $b, \Sigma, \mu$ depend smoothly on $x$. The third point is equivalent to saying that the solution $(X_t)_{t\ge0}$ to the SDE exists almost surely on infinite time interval (see [Schilling (1998)](https://link.springer.com/article/10.1023/A:1008664419747)), which is the case for the perturbed gradient flow with respect to second differentiable loss.

---

> > ### Comment · Reviewer_bLX6 · 2022-08-09
> > **Thank you for your response**
> >
> > I think that adding this discussion to the paper will improve the clarity of the results.

---

### Official Review · Reviewer_Za8N · 2022-07-10

**Rating:** 6
**Confidence:** 2
**Soundness:** 2 fair
**Presentation:** 4 excellent
**Contribution:** 2 fair

**Summary:**

Stochastic gradient descent often generalizes better than deterministic gradient descent.  Motivated by the intuition that _chaotic_ deterministic systems act like stochastic systems, this paper proposes to train neural networks using a chaotic version of deterministic gradient descent.  In particular, the idea is to inject perturbations at each step which are driven by a chaotic dynamical process.  The authors argue that this approach may help deterministic gradient descent generalize better.

Theoretically, the authors:
 1. prove a generalization bound for their method that leverages recent generalization bounds for heavy-tailed SGD
 2. prove that their method approximately follows gradient flow on a regularized loss functional which implicitly penalizes the Hessian trace.

Empirically, the authors:
1. Demonstrate that on a toy problem (the "widening valley"), their method finds a flat minimum even though gradient descent with a small learning rate finds the sharp minimum that it was initialized near.
2.  Demonstrate that on a UCI regression problem, there exists a hyperparameter setting for their method (which has three hyperparameters)  and a corresponding hyperparameter setting for deterministic gradient descent (which has one parameter, the step size) where their method achieves lower RMSE than deterministic gradient descent.
3.  Demonstrate that on CIFAR-10 classification with a ResNet-18, there exists a hyperparameter setting for their method (which has three hyperparameters) and a corresponding hyperparameter setting for deterministic gradient descent (which has one hyperparameter, the step size) for which their method achieves better test accuracy than deterministic gradient descent.


**Questions:**

If you tune the step size for the baseline of deterministic GD, does the proposed method still work better?

**Limitations:**

Yes

**Strengths And Weaknesses:**

I don't understand most of the theory in this paper, and I will leave evaluation of the theory to the other reviewers.

Regarding the impact on deep learning practice,
 1. I am not convinced that the method actually does robustly generalize better than full-batch gradient descent.  It does not seem that the authors have tuned the step size for the gradient descent baseline, whereas the authors _have_ tuned the three hyperparameters for their method.  If one tunes the hyperparameters for both methods on a validation set and evaluates the performance of the optimal hyperparameters on a separate test set, does the proposed method actually generalize better than deterministic GD?
 2. to whatever extent the method does robustly generalize better than full-batch gradient descent, I would be surprised if this is explained by the theory in the paper.  Concretely, I would wager that there are a huge number of schemes for injecting perturbations into gradient descent for which one can get a small boost in test accuracy.  I have a hard time believing that the fact that this particular scheme "homogenizes to a limiting SDE driven by a heavy-tailed Levy stable process" is relevant.  Note that the overall message of the Geiping et al paper is that there is nothing special about randomness that promotes generalization.  Do the authors agree with that message?   Please correct me if I am wrong, but the premise of this submission seems to be that there is something special about randomness that promotes good generalization, and that this magic can be mimicked by injecting chaos into the algorithm, since chaos $\approx$ randomness.

Minor point: in motivating this study, the authors write in their introduction
> Contrary to the wisdom of optimization theory, which typically suggests using smaller learning rates, the fact that large learning rates often provide better generalization performance in modern nonconvex optimization problems has opened up several research directions. In this context, [CKL+21]  showed that large learning rates drive the GD dynamics towards the ‘edge of stability’, meaning that the learning rate should be chosen large enough to make the algorithm diverge.

The paper CKL+21 is not about generalization or large learning rates.

---

> ### Author Response · Authors · 2022-08-02
> **Thank you for the review and the questions**
>
> We thank the reviewer for the insightful questions. Before directly answering the questions, we would like to clarify one point, which might help resolve the concerns.
>
> Depending on the hyperparameters of GD, the behavior of the  algorithm might significantly differ. For instance, when the step-size is chosen larger than some threshold, the algorithm might exhibit a chaotic behavior. By using the analogy provided by the reviewer, i.e., "chaos $\approx$ randomness", even in the setting of Geiping et al., the performance improvement might be brought by such *"implicit randomness"*, which arises from total determinism.
>
> In this sense, we believe that the take home message indicated by the reviewer, that is "There is nothing special about randomness that promotes generalization." shall be revised to "There is nothing special about *explicit* randomness that promotes generalization."; the implicit stochasticity generated by GD might still be crucial for generalization, and our goal in this paper is to take another step towards demystifying this case. (Note that there is no explicit randomness in our framework, the system is totally deterministic).
>
> Indeed, a similar statement has already been made in Geiping et al (arxiv 2109.14119, Section B7, Page 25). Let us cite the relevant part for the convenience of the reviewer:
>
> "We would like to point out that while our results show that stochastic mini-batching (or even nonstochastic minibatching) in gradient descent is not necessary to achieve state-of-the-art generalization behavior, this does not entirely rule out stochastic modeling of the behavior of GD for deep neural networks as proposed in works such as Chaudhari and Soatto (2018); Kunin et al. (2021) and
> Simsekli et al. (2019). Even a full-batch gradient descent algorithm could potentially exhibit chaotic
> behavior on the loss surface of deep neural networks (Kong and Tao, 2020), which could be modelled
> by statistical techniques. In this work, we can make no statement about whether chaotic behavior
> exists for these examples of gradient descent and whether it has an impact on model performance."
>
> As we mentioned in our response to Reviewer TqfM, we believe that the setting considered in Geiping et al. might be analyzed in our theoretical framework with certain modifications, and our theoretical framework can provide useful tools for identifying the role of implicit randomness in deterministic optimization algorithms. Due to the current density of our paper, we leave it as a future work.
>
> Our responses to the two main questions are as follows:
>
> 1. We would like to note that, we did not tune the step-size for any of the algorithms. Since our generalization bound applies to the asymptotic SDE, which is obtained when the step-size goes to zero, in order to stay close to the theory, we chose a small enough step-size to be both not far from the continuous dynamics, and large-enough so that the algorithm converges in a reasonable amount of time. That being said, we have tried a range of step-sizes for both algorithms, and we observed that the proposed scheme consistently outperforms vanilla GD for smaller step-sizes. Whereas if we use a large step-size, both algorithms perform similarly. In this regime, vanilla GD as well potentially emits a chaotic behavior, which might also indicate the importance of the "implicit randomness". However, as we indicated in the introduction, this regime is not easily controllable (cf. Line 49), and our purpose is to introduce a controlled chaotic component, with a clear theoretical understanding. In this respect, we believe that the proposed scheme achieves its goal. We will mention this point more clearly in the paper.
>
> 2. Above, we have responded to the question regarding the main message of Geiping et al.
>
> On the comment regarding the relevance of the heavy-tailed SDEs, we would like to underline that, recently, it has been both empirically and theoretically illustrated that heavy-tails might have an important impact on the generalization error in *stochastic* optimization. See for instance:
>
> Mahoney, Michael, and Charles Martin. "Traditional and heavy tailed self regularization in neural network models." International Conference on Machine Learning. PMLR, 2019.
>
> Simsekli, Umut, et al. "Hausdorff dimension, heavy tails, and generalization in neural networks." Advances in Neural Information Processing Systems 33 (2020): 5138-5151.
>
> Our work shows that heavy-tails might even be observed in deterministic optimization. Given that the link between heavy-tails and generalization error does exist in the literature and has already been investigated to a certain extent, we do believe that the limiting heavy-tailed SDE and its link to generalization are indeed relevant and well-grounded.  We will clarify this point in the paper.

---

### Official Review · Reviewer_TqfM · 2022-07-11

**Rating:** 4
**Confidence:** 3
**Soundness:** 3 good
**Presentation:** 3 good
**Contribution:** 3 good

**Summary:**

This paper introduces a multiscale perturbed gradient descent (MPGD) framework for global optimization, which aims to improve the generalization ability of the resulting predictive model.

**Questions:**

1. What is advantage of the proposed method compared to SGD?   It seems that the proposed method works with full data at each iteration, which can be much slower than SGD.

2. The performance of the proposed method is not clearly illustrated. The SGD and some regularized SGD algorithms should be used as the baseline in numerical experiments.

**Limitations:**

Yes.

**Strengths And Weaknesses:**

The theoretical result makes sense to me, although I did not check the proof carefully.  Compared to stochastic gradient descent (SGD), the advantage of the proposed method is not clearly addressed.

---

> ### Author Response · Authors · 2022-08-02
> **Thank you for reviewing the paper**
>
> We thank the reviewer for their time invested in our paper. Regarding the reviewer's questions, we suspect that there has been a misunderstanding regarding the scope and purpose of our paper. We believe we can clarify the situation and we hope that the reviewer could reconsider their score based on our explanations.
>
> In the recent years, there has been an increasing interest in understanding the role of stochasticity in non-convex optimization. While SGD is computationally more efficient and often provides improved predictive performance empirically (as discussed for the CIFAR-10 task in the paper), lately it has been empirically illustrated that under suitable learning rate schedules and regularization tricks, GD can achieve similar results to the ones of SGD (see Geiping et al., 2021). In this setting, where unusually large learning rates are chosen with exotic learning rate schedules, a natural question is whether or not GD exhibits a "chaotic behavior", which might make GD behave somehow "stochastically", even though the algorithm is *fully deterministic*.
>
> This question was partially answered in Kong and Tao, 2020, where the authors showed that, for large learning rates deterministic GD becomes *chaotic* and behaves similar to a stochastic optimization algorithm in a specific setting. Hence, one might suspect that the performance improvement obtained in this setting can be partially explained by this chaotic-stochastic behavior.
>
>
>
> Our paper contributes to this line of research, where our primary goal is to understand the behavior of deterministic gradient descent in a chaotic setting. In this respect, we shall underline that, at this stage, our goal is *not* to develop a competitive algorithm that would outperform SGD, but rather to provide a solid theoretical analysis for how stochastic behavior can emerge from GD with suitable *deterministic* components, as well as their implication in terms of generalization error. We have introduced a novel theoretical framework for analyzing GD by drawing non-trivial connections to rough paths theory and statistical learning theory. We believe that our contributions should be evaluated within this context.
>
>
> Nevertheless, we believe that the setting considered in Geiping et al. (which provided competitive performances compared to SGD), can be analyzed in our theoretical framework with certain modifications. In this respect, our framework has the potential of giving rise to competitive algorithms; however, to ensure that the present paper stays focused, we left out such investigation from the present paper. We believe that such study is a natural next step and deserves a separate paper on its own.

---

### Official Review · Reviewer_AQWb · 2022-07-11

**Rating:** 8
**Confidence:** 2
**Soundness:** 4 excellent
**Presentation:** 4 excellent
**Contribution:** 4 excellent

**Summary:**

This paper proposes a modified gradient descent algorithm by adding a chaotic component in the algorithm in a careful manner. The authors relate the algorithm to its SDE limit with a heavy-tailed Levy process when the step size goes to zero, and derive its generalization error bound. They also show that the chaotic term has a desired regularization effect that penalizes the Hession matrix, which is then illustrated by empirics.

**Questions:**

1. For the generalization bound, does the propose algorithm benefit or loose anything, in terms of convergence rate, when the loss function is convex or strongly convex?

2. Does the algorithm parameters, $\mu$, $\sigma$ play an role in the generalization bound? In either case, it is worth discussion.

3. Besides the numerical experiments, can you provide some qualitative result that identify regimes where the proposed algorithm chaotic regularized GD works better than GD?



**Limitations:**

The authors adequately addressed the limitations and potential negative societal impact.

**Strengths And Weaknesses:**

**Strength**

1. This is a well-written paper. Although I am not an expert in this field, I enjoy reading the paper. The main message is clear, and the authors pay close attention to mathematical rigor.

2. The paper proposes a new variant of gradient descent algorithm and provides a comprehensive analysis on its SDE limit, the generalization bound and the regularization effect, and also demonstrate via numerical experiments.

**Weakness**

1. The paper contains a lot of technical results to me. Its readability may be improved by supporting some specialized examples and contrast with vanilla Gradient Descent.

---

> ### Author Response · Authors · 2022-08-02
> **Thank you for careful reading of the paper**
>
> We thank the reviewer for careful reading of the paper as well as the encouraging comments and suggestions. We will make sure to use the additional content page for the camera-ready version to expand on the technical results and improve the readability.
>
> - Regarding the question "Does the proposed algorithm benefit or lose anything, in terms of convergence rate, when the loss function is convex or strongly convex?": Our theorem holds under quite general assumptions on the loss function but constant (in terms of the GD iterations $k$) step-size $\eta=1/m.$ We suspect that certain regularity assumptions on the loss would help accelerate the convergence; however, within our current theoretical framework, the convexity or strong convexity assumptions would not directly provide a gain or loss.
>
> - Regarding the question "Do the algorithm parameters play an role in the generalization bound?": The algorithm parameters interact with the bound via the constant terms $K_1$,  $K_2$, and the mutual information terms $I_1$ and $I_\infty$. However, unfortunately their dependence is quite implicit and we are not able to provide quantitative estimations for these constants at this point. We agree that this aspect requires more discussion and we will clarify this point in the paper as suggested.
>
>
> - Regarding the question "Can you provide some qualitative result that identify regimes where the proposed algorithm chaotic regularized GD works better than GD?": We observe that in the weak perturbation regime  and for non-convex losses, MPGD is more beneficial than GD in the sense that it reduces the Hessian of the loss, thereby promoting flatter minima. Flat minima have been widely believed to lead to improved generalization in neural networks. Hence, in order to observe the benefits of our scheme, we suspect that the optimization problem should be non-convex and local minima should have different curvatures. We will add a comment on this subject in the paper.

---

### Meta-Review · Area_Chair_WYCs · 2022-08-21

**Recommendation:** Accept
**Confidence:** Less certain

**Metareview:**

A recent line of work on the role of stochasticity in ML suggests that variants of GD which use non-traditional step size schedules (a) may perform relatively well in certain settings (b) due to an implicit chaotic behavior. The present paper studies a variant of GD (MPGD) augmented with an explicit chaotic component, implemented by means of an external deterministic dynamical system, as a theoretical model for investigating these hypotheses. Recent results are shown to imply generalization bounds for the limiting stochastic process. Numerical results are provided for comparing the performance of MPGD to existing methods.

The reviewers have generally found the use of a GD variant with an explicit chaotic term, as well as the proposed analytic framework, interesting and appreciated the clarity and rigor of the results given in the paper. In later discussions, concerns regarding the relevance of the theoretical model to (a) and (b) above were raised by the reviewers, questioning more broadly the significance of MPGD and the respective limiting SDE to the general understanding of SGD/GD. All in all, I think this is a reasonable paper to accept if there is room. The authors are encouraged to revise the paper according to the important feedback given by the reviewers.





**Award:**

No

---

### Decision · Program_Chairs · 2022-09-14

Accept